# Sweet Basil between the Soul and the Table—Transformation of Traditional Knowledge on *Ocimum basilicum* L. in Bulgaria

**DOI:** 10.3390/plants12152771

**Published:** 2023-07-26

**Authors:** Teodora Ivanova, Yulia Bosseva, Mihail Chervenkov, Dessislava Dimitrova

**Affiliations:** 1Department of Plant and Fungal Diversity and Resources, Institute of Biodiversity and Ecosystem Research, Bulgarian Academy of Sciences, 1113 Sofia, Bulgaria; julibos@abv.bg (Y.B.); vdmchervenkov@abv.bg (M.C.); dessidim3010@gmail.com (D.D.); 2Faculty of Veterinary Medicine, University of Forestry, 1797 Sofia, Bulgaria

**Keywords:** species identification, plant blindness, awareness disparity, food plants, essential oils, local landraces, plant varieties, medicinal plants, linalool

## Abstract

The study tracks the utilization of *Ocimum basilicum* L. (sweet basil)—a garden plant popular for its ritual and ornamental value in the past, that is currently applied in various forms and ways as medicine, food, insect repellent, etc.—in Bulgaria. Previous data for Bulgarian rural home gardens showed a significant number of preserved local landraces; however, it remained unclear how people perceive the large varietal diversity of this species and how the traditions related to its use are preserved. We combined a literature review on the cultural value of sweet basil and the breeding of local genetic resources with an online questionnaire, directed to adult laypeople, that sought to access different aspects of past (recalled) and present use and related knowledge. The identification skills of the participants were tested using images of local plant landraces and foreign varieties. Responses from 220 participants showed that potted “Genovese”-type individual was most frequently identified as sweet basil (89.9%), followed by two examples of local landraces in flower. Participants who grow sweet basil or used it in more varied ways had significantly better identification skills. *Ocimum basilicum* was most frequently reported as food, while ritual/symbolic use was preserved while devalued during the Communism regime (1945–1989). Food and religious uses were negatively associated in the past, but presently, the tendency is completely reversed. Preferences for the informal exchange of seeds and seed-saving practices were discussed.

## 1. Introduction

The diverse profile of plant–people interactions reflects socio-economic development and cultural backgrounds, both currently rapidly changing as a consequence of global climate and political shifts [1,2,3]. Plant genetic resources developed by local communities (i.e., landraces, heirloom, old or farmers’ varieties) gain more and more attention in order to support the adaptation potential and sustainability of agriculture [4,5,6]. However, farmers and communities that maintain them continue to be under constant pressure to conform with shifts in market requirements and agricultural and land policies, as well as to cope with the consequences of growing rural area depopulation and climate change while upholding their cultural preferences [5]. Despite the arduous endeavors of many local and international initiatives, combined in situ and ex situ conservation efforts to preserve these resources currently remain “unplanned and uncoordinated”, struggling to attract enough attention to much-needed actions that concern both crops and their wild relatives [6,7,8,9]. The majority of edible plants have other uses, usually more than one, e.g., over 70% of them are also used for medicinal purposes [10]. It is generally accepted that societies/communities which are more reliant on natural resources are more deeply involved with nature [11,12]. However, these societies show less attention to endangered (often rare) wild plants compared to the interest toward common plants and those used for food, medicine, ritual purposes, etc. [11,13]. Wealthier countries, on the other hand, have been deemed less and less connected with plants mainly due to urbanization, modernization, and industrialization, especially in agriculture and forestry [14]. Yet, it remains debatable what kind of interventions would overcome plant blindness (awareness disparity) as the goals and outcomes of biological education across societies vary considerably and most of the studies on species literacy and plant identification skills among different age groups and professions are disproportionately centered on university students, mostly in Northern America and some European countries (e.g., Scandinavia, Spain, Great Britain, Baltic countries, etc.) [3,15,16,17,18,19,20].

Numerous studies present data on the preserved cultural importance of plants and their communities, habitats, plant-related landmarks, landscapes, and various services they provide [21,22,23,24,25,26,27,28,29,30,31]. Although it is not easy to track the origins and transformation of the related (local) knowledge, it remains crucial for the prevention of biodiversity and habitat loss and adaptation to climate change [32,33,34,35]. Traditional knowledge was also shown to support the implementation of (agro) biodiversity conservation measures, and in recent years, the research data in the field have been increasing [36,37,38]. Thus, it is important to explore the motivations related to the preservation and persistence of culturally relevant (plant) biodiversity and associated knowledge [39,40,41,42,43]. The transformation of plant-related traditional knowledge in Europe has been reported mainly with respect to its abandonment and its potential role in changing attitudes towards the responsible use of biodiversity in local entrepreneurial activities [1,39,44,45,46,47,48,49]. Research on past and present modes of utilization as well as factors that influence the transmission of this knowledge is focused more on wild plants [10,50,51,52], and currently, only a few European countries hold systematic knowledge on traditional landraces [53,54]. Recent projects that target the transformation and preservation of biocultural knowledge in Europe also stress its homogenization, especially in regions like the Balkans and Eastern Europe, where prolonged sociopolitical instability during the early 20th century and the following Communist internationalistic policies severely affected rural livelihoods and intergenerational cultural transmission [55,56,57,58]. Hence, some authors even consider the domestication of edible plants, otherwise collected from wild populations, as an attempt to support food-insecure communities [59].

We track the biocultural fate of *Ocimum basilicum* L. (Lamiaceae) in Bulgaria as a representative example of the transformation of the knowledge and importance (meanings) related to this popular and easily cultivated plant both in pots and in gardens. The genus *Ocimum* comprises more than 70 species that are native to tropical and subtropical regions of Africa, Asia, and Central and South America, many of which have been used in various ways by humans since Antiquity [60,61]. Ancient authors in the Mediterranean region mentioned it as an early, fast-germinating leafy crop (seasoning herb) and medicinal plant, important both in veterinary practice (as purgative for cattle) [62,63] and as a human remedy for burred eyesight, respiratory problems, depression, swellings, scorpion stings, snake bites, etc. [63,64,65]. Currently, diverse biological activities of *Ocimum* extracts and essential oils have found application in a wide spectrum of medicines and food additives [66,67,68]. Christianity and Hinduism are two major religions that have held basils as plants with sacred symbolic and ritual value—in the former, it is *Ocimum basilicum* L. and in the latter, *Ocimum tenuiflorum* L. (syn. *Ocimum sanctum* L.). That was a significant factor in their secondary distribution around the world [60,69,70]. Along with its religious meanings, *O. basilicum* (sweet basil) was considered more as a medicinal (incl. aromatic) and decorative plant, largely cultivated in Bulgarian gardens, at least until World War II [71,72]. The all-curative perception of *O. basilicum* could be related to the homophony of its Bulgarian common name *bosilek* (*бoсилек*, in Greek *βασιλικός, royal*) with *lȇk* (*лѐк*, from Proto-Slavic, *medicine*). Similarly, *bosilek* is also incorporated in the Bulgarian common names of some wild medicinal plants from the Lamiaceae family like shepherd’s basil (*ovcharski bosilek*) for *Thymus* sp. and *Origanum vulgare* L., horse’s basil (*konski bosilek*) for *Salvia pratensis* L. and *Mentha longifolia* (L.) L., and wild basil (*div bosilek*) for *Clinopodium suaveolens* (Sm.) Kuntze and *Mentha longifolia* (L.) L., which could lead to the misnaming and mislabeling of these plants and related drugs [73]. Some Bulgarian ethnographic sources denote the *O. basilicum* varieties with “larger flowers” as *гръцки бoсилек* or “Greek” basil [74], and their broad-leaf counterparts (e.g., “Italiano Classico”, “Genovese”, etc., cultivars), informally called “Italian” basil, are ubiquitously available on the market for hobby gardeners since the import of foreign seeds was relieved after accession to the European Union in 2007. Although numerous nameless landraces of *O. basilicum* were found grown in Bulgarian rural gardens, only 39 accessions of local landraces of the species have been listed in Bulgarian National Inventory data in the EURISCO Database, most of which were collected after 2007 [75,76]. Sweet basil was reported recently for landrace hotspots in Italy and Greece; however, about half of the accessions in European gene banks (a total of 816) are stored in Germany, Croatia, and Ukraine [75,77].

The aim of the present study was to discern which plant characteristics, denomination(s), and/or use modalities are significant (focusing on laypeople) for affiliating and discerning *O. basilicum* as culturally important, currently and in the past. Cultivation for personal use is discussed as a factor in the preservation of local forms/landraces.

## 2. Results and Discussion

### 2.1. Cultural Context and Historical Relevance

The ritual use of *O. basilicum* in Europe was recorded both for Orthodox and Catholic Christian denominations, with strong connotations to marital and burial rites, the latter related to basil’s denotations in the resurrection of Christ [60,78]. Hence, sweet basil was incorporated into folklore and could be found in ritual settings and decorations of people, icons, and buildings [71,79,80,81,82]. Being used in a variety of rituals and ceremonies, *O. basilicum* was popularly cultivated in home gardens and is quite frequently mentioned in Bulgarian folklore, and in that of other Balkan countries [71,83,84,85,86]. Vakarelski (1946) mentioned it as one of the important ritual plants used by the Orthodox Church in his instruction manual for field ethnographical studies in the “Plant world in Spiritual culture” section [87]. The symbolism related to sweet basil is popular in several Slavic countries, but it could be related rather to Orthodox Christianity than to some pre-Christian roots, as it is not so frequently reported for Central European countries where Catholicism is prevalent [79,80,88]. Its function in blessing bunches of Orthodox priests was considered part of the primitive/folk Christian ritual practices in Bulgaria that sought to connect earlier rituals with the Christian faith [89]. In the Kyustendil region (Southwestern Bulgaria), there is a preserved practice to celebrate the wheat harvest and the intergenerational transfer of knowledge related to breadmaking that is intertwined with the Orthodox Lifting of the Breads (Panagia) during the Dormition of the Mother of God commemorations on 15 August [90,91]. Flowering sweet basil together with other seasonal ornamentals like *Geranium macrorrhizum* L., *Chrysanthemum indicum* L., *Zinnia elegans* Jacq., *Tagetes erecta* L., etc., are used for the decoration of sacred bread and icons (Figure 1). Sweet basil oil is also used as a non-essential ingredient of the Great Chrism made by Bulgarian and Romanian Orthodox Churches [92,93].

Religious/ritual use of sweet basil was reduced during Communism (1944–1989), parallel to the oppression of the Orthodox Church along with all other denominations [95]. State antireligious policies prohibited public displays of faith and participation in rituals, which minimized and forced all rites and related ritual elements into private spaces [96]. Thus, festive and ritual decorations and ritual objects that would incorporate sweet basil and other symbolic plants became obsolete in public contexts. This was also reflected in floriculture at the time, when *Ocimum* was described as “a plant introduced before the Liberation (from Ottoman Empire) … of low decorative value, but still very fragrant” [97].

The medicinal properties of sweet basil are widely popular and related to its ritual functions in Bulgarian folk and modern medicine. Aside from the oral administration of infusions, extracts, and medicinal wines made of the herbage and seeds and fumigations for various ailments, in Bulgarian folk medicine, sweet basil branches are used to sprinkle other herbal infusions to improve their curative strength, as well as for cauterizations, and worn as an apotropaic bunch [74,98]. Sweet basil is also considered a fortune/health-bringing object, especially in marital and midwifery rituals (related also to the health of the mother and the newborn) [83,99,100]. The flowering herbage and seeds are mentioned in numerous preparations in all major professionally written phytotherapy textbooks and herbals, as well as in the multivolume compendium of folk prescriptions—Peter Dimkov’s Bulgarian folk medicine (1926–1939) [101,102,103,104,105]. Most of the biological activities mentioned there (e.g., antibacterial, spasmolytic, carminative, orexigenic, anti-inflammatory, antitussive, anticonvulsant, relaxant, broncholytic, etc.) were found to be related to the constituents in the essential oil. It was valued in Indian, Chinese, and other Asian medical practices and further translated into modern medicine for the treatment of colds, coughs, urinary complications, earaches, menstrual irregularities, arthritis, and viral, fungal, and bacterial infections, to name a few [106,107]. Interestingly, the historical ethnopharmacological data from other Balkan countries showed limited use of *O. basilicum* [108,109,110]. Due to the broad profile of volatiles in the essential oil, like linalool, methyl chavicol, eugenol, bergamotene, and methyl cinnamate, *O. basilicum* varieties are often identified both by their morphological traits and as chemotypes [111,112]. Varieties considered most suitable for culinary purposes were those with linalool content ranging between 19% and 38% [113]. The typical Bulgarian chemotype belongs to the European group but was reported to contain high linalool levels (50–70%). Some authors report also methyl cinnamate as a characteristic compound [114,115,116,117,118]. Indeed, with the growth of the essential oil industry in Bulgaria after the 1940s, sweet basil had become one of the largely explored cash crops. The production peak was in the late 1980s and early 1990s, with Bulgaria being the largest European producer of sweet basil oil, known also as Bulgarian basil oil [115]. Professionally selected Bulgarian cultivars of *O. basilicum* (“Trakia”, “Jubileen”, “Mesten”, and others), developed in the 1960s–1980s, were also driven by the extraction of essential oils with linalool as the main prevailing component, and were sought for industrial cultivation, even outside Bulgaria [119,120,121]. Linalool is largely known for its use in perfumery, cosmetics, and household products and is much less popular in cooking, which could explain the reluctance of some seniors to consume it, together with the plant’s connotations to funerals and death in some regions in Bulgaria [122,123]. Large quantities of linalool in Bulgarian sweet basils would easily explain the complete absence of sweet basil from earlier cookbooks and Communist catering recipe compendiums, not only in Bulgarian traditional dishes but also in Italian-style ones [124,125,126,127,128,129,130,131]. Some authors mentioned it, specifically *Ocimum basilicum* f. *minimum*, as a food plant used in sausages, pickles, and stews, but without any recipes provided [72,101,132]. On the other hand, currently, popular TV chefs are promoting the use of sweet basil not only as a part of foreign cuisines but also as an ingredient in some traditional dishes like beans stew in a clay pot and *patatnik*, a dish originating from Rhodopi Mts., made of grated potatoes with cheese or veal and/or mutton [133,134].

### 2.2. Online Questionnaire

Mixed closed- and open-ended responses of 220 adults were collected though an anonymous online questionnaire (Table 1). Most of the participants belonged to age groups of 21–40 (44.7%) and 41–60 years of age (43.3%), which correlated well with the age profile of Internet users in Bulgaria [135]. The prevalence of women (86%) could be associated with the higher involvement of women in sharing and general interest in factual plant-related information and also in gardening and cooking [136,137,138,139]. Similarly, the prevailing educational level was third level (university/college); however, there was little involvement from holders of biology and/or agriculture diplomas (under 15%). Participants who spent their formative years and started their compulsory education (6–8 years of age) before the end of Communism in Bulgaria were 51% of the sample (coded hereafter CDC, Childhood during Communism, as opposed to CDD, Childhood during Democracy).

All participants identified at least one image of *O. basilicum*, and the negative controls, *Mentha* sp. (presented by its owner as “Greek basil”, bought from Greece) and *Salvia officinalis*, were mislabeled as sweet basil by less than 5% of the participants (both 3.7%, *p* < 0.01, Chi-Square test, Figure 2). *Ocimum basilicum* is known for its variable morphometric characteristics that depend on the genotype and the pedoclimatic conditions, as reflected in the numerous intraspecific taxa, many of which are currently transferred to synonymy [61,140]. This could explain why only two of six presented pictures of *O. basilicum* were clearly identified as sweet basil, namely a potted “Genovese” variety with large convex leaves (89.8%, *p* < 0.01) and a flowering individual of a local garden landrace from Ivaylovgrad (South Eastern Bulgaria) (58.6%, *p* < 0.05). Contrastingly, the other image presenting flowering *O. basilicum* (garden landrace from Samuilovo village, Southwestern Bulgaria) was positively identified only by 39.1% of the participants (*p* < 0.01), supposedly due to the lack of close-up details of flowers and leaves. The least identifiable O. *basilicum* was a dwarf “Globe” variety, internationally marketed as “Greek” basil (5.1%, *p* < 0.01) [141]. The specialized education and/or gardening hobby, age, or sex were found non-significant for plant identification when the study sample was taken as a whole (*p* > 0.05, Kruskal–Wallis and Mann–Whitney tests, data not shown).

The absolute number of CDC participants who identified the presented taxa was higher than that of CDD participants, and only for one variety and one landrace was the difference in favor of CDD, by only one participant (Table 2). When comparing the accuracy of CDC and CDD groups, there was a significant difference only for the “Genovese” variety (*p* = 0.024, Fisher’s exact test). Identification skills and the time of the participant’s childhood were significantly associated only for the “Genovese” variety and *Mentha* sp. (φ_c_ > 1, small association).

Identification accuracy was associated with the direct availability of *O. basilicum* in participants’ homes (in pots) or in a home garden. The lack of interest and/or opportunities to cultivate was associated with a significantly lower ability to correctly identify most of the presented varieties/landraces, except for the least identifiable dwarf variety (Figure 3). Having potted sweet basil plants was found important for the most identifiable “Genovese” variety and the two landraces pictured in the flowering stage but not relevant for the rest of the cases (*p* < 0.05, Fisher’s exact test). 

Ready potted plants from the market were the preferred way to obtain plants at home (21.1%, N = 147), and only 6.7% of the participants had saved seeds for more than 10 years (*p* = 0.001, Chi-Square test, Figure 4a). Plants and seeds were procured most frequently through non-formal exchange and purchasing seeds from Bulgarian producers, an approach that could not guarantee the quality and origin of the seeds (Figure 4b). Vegetable plots and amateur greenhouses were found very popular among Bulgarians and home gardens are frequently used for semi-subsistence agriculture not only in the rural but also in urban and peri-urban areas, which motivates the procuring of seeds and other planting materials of various origins [123,142]. Still, non-formal exchange and seed saving could misleadingly reassure laypeople about the authenticity of the landraces and varieties due to cross-pollination between varieties and species in the genus *Ocimum* [143]. 

About half of the participants correctly identified no more than two pictures of *O. basilicum* varieties/landraces, with 35.4% correctly identifying only one (Figure 5a,b). Although the number of CDC and CDD participants who identified three or more sweet basil varieties/landraces was similar, 10% more CDC participants correctly identified at least two images. 

Most of our participants (75.8%, *p* < 0.001, Fisher’s exact test) were previously aware of different *O. basilicum* varieties, with 45.7% certain in their ability to discriminate among them, and 32.6% who were familiar, but not sure they could discriminate so easily. A significant number of CDC participants (83.2%) were more certain about their knowledge of different varieties than CDD participants (70.8%) (*p* = 0.036, Fisher’s exact test). 

For younger participants (CDD), cultivation, both in gardens and in pots, was equally important for the number of identified *O. basilicum* images, whereas for CDC, only experience in gardening was a significant factor (*p* < 0.001, Fisher’s exact test). This could be related to the low percentage of CDD participants involved in growing sweet basil in home gardens, which was considerably lower compared to CDC participants (31.1% to 45.4%, respectively). Age and more intensive contact with nature (like living in rural areas) were shown as important predictors of plant identification skills, as these experiences allow the accumulation of additional knowledge throughout the years [15,144]. Prior conditioning/education was also shown to improve species identification in general, but within our sample, such calculations would be misleading due to the prevailing participation of people without specialized higher education, and species identification is not emphasized enough in general primary and high school education in Bulgaria [15,145,146].

It was shown that various factors influence plant identification skills, i.e., local plant richness, demographic and socioeconomic factors, proficiency/professional involvement, source(s) of information, personal interests, etc. [16,147,148,149,150,151]. Previous studies argued that the presentation of real plants (cuttings, potted plants, etc.) results in better identification compared to images as they provide additional clues for the interviewees; however, this is not always possible and/or feasible [15,152]. In the majority of those cases, participants are bound to free-list taxa (majority of ethnobotanical studies) and/or to recognize sampled individuals/images of plants on a species level (together with mammals, insects, birds, etc., for species identification research), often combining species examples with drastically dissimilar features like different lifeforms, phenological stages, etc. Additionally, flowering plants were usually presented in the flowering stage as it is considered the most representative and attention-attracting [15,153]. In this study, over 96% of the participants discerned with great certainty the three species (mint, sweet basil, and common sage), among which *Mentha* sp. and *O. basilicum* were in the flowering stage. This supports the notion that Lamiaceae taxa are among the most popular medicinal and aromatic plants (wild and cultivated) to Bulgarians [76,154] and implied that identification relies on various aspects, e.g., in what phenological stage or form the specific taxa is most frequently seen and/or used. 

Less than half of all participants agreed to answer the question of which of the presented images depicts “Bulgarian” or “*nash(enski)*” (ours) sweet basil. The landrace from Ivaylovgrad (picture B) gathered the most positive responses (28.57%, N = 98), followed by the “none of the presented” category and landrace from Samuilovo (image C) (27. 55% and 16. 33%, respectively, *p* > 0.05, Chi-Square test, Figure 6). The “Bush” variety was most frequently identified as the “Greek” variety (image D, 59. 06%, N = 149, *p* < 0.001) and the “Genovese” as “Italian” (image A, 73.29%, N = 161, *p* < 0.001). Remarkably, all participants were reluctant to assign specific use to any of the three country-bound identities, which was opposite to data on the Italian diaspora in Romania who discriminate between “their” (Romanian) and “our” (Italian) sweet basil solely according to its ritual and culinary use, respectively [78]. However, further exploration of the identification–usage nexus showed that high positive identification of the “Genovese” variety was reflected in the use frequencies.

The presence of flowers was expected to be helpful for the identification, due to the traditional symbolic value and popular use of *O. basilicum* as a medicinal and ornamental plant. However, identification accuracy was significantly associated with most of the current uses reported by participants but less with those they recall from their childhood memories (Figure 7). The food category was most frequent both for current and childhood years (83.3% and 44.7%, respectively, Table 3). However, this increase of about twofold in the number of participants using *O. basilicum* as a food plant was not associated with improved plant identification skills (*p* = 0.130, Fisher’s exact test). The majority of the participants who used sweet basil as a culinary herb, both currently and in the past, tended to recognize one to three varieties/landraces, among which the “Genovese” variety prevailed. On the opposite side were ornamental (*p* = 0.012), medicinal (*p* = 0.002), and religious/ritual use (*p* < 0.001), for which the number of participants who correctly affiliated images peaked at four. Interestingly, for the latter, only past use was significantly associated with identification accuracy (*p* = 0.007). 

Data on the consumption of sweet basil showed that participants born during the Communist era use it more frequently in dishes both of Bulgarian and international cuisines compared to CDD participants (Fisher’s exact test *p* = 0.019, φ_c_ = 0.214 (small association)). Very few of the CDD participants reported usage of *O. basilicum* only in Bulgarian dishes (12.6% to 28. 5% for CDC, Figure 8), which requires further research as there were no available data on the prevalence of consumption of home-cooked meals and cooking skills among this group in Bulgaria. Additionally, it was previously shown that even for some recipes that are explicitly considered traditional, the way of preparation is more important for Bulgarians than the ingredients used [155].

Religious/ritual use was the category in which minimal change was observed between use frequencies in the past and the present—about 36% of the participants, with about twice as many CDC continuously using *O. basilicum* in comparison to CDD ones (*p* ≤ 0.01, Table 3 and Table 4). Only 42 participants considered themselves as observing religious practices involving sweet basil, with CDC prevailing (70%, *p* = 0.039, Fisher’s exact test, φ_c_ = 0.251). Nearly half of the participants who replied to the question on the rituals that involve sweet basil were not sure about its role (48.6%, N = 109), and the rest were divided between blessings (for health, inauguration, protection, etc.—27.5%) and funeral rites (23.9%). In all other categories, there was an increase in the reported use frequency, with medicinal and aromatic use growing from 34.4% to 49.3% and insect repellent use rising from 22.3% to 34%. For the latter, together with food, past and present use were not significantly related. The seniors (CDC) who started to consume sweet basil in their adult years constituted the major portion of participants who contributed to the significant increase in the food and medicinal categories (*p* = 0.05 and *p* = 0.01, respectively, Table 4). While the data for the food category could be related to the current availability of new imported varieties, the increase in medicinal use is more likely due to the persistence of traditional medicinal practices and lower prices of herbal medicines, shown to be important factors in late senior years, especially for women [156,157]. Intergenerational knowledge transfer was also shown to be important in the persistence of traditional medicinal practices [158]. 

Numerous books published in Bulgaria that cite folk medicine indicate that traditional knowledge was not neglected but favored by the state during Communism, in contrast to religion. Hence, the interest not only in traditional/“old” knowledge but also in the traditional medicine of other cultures was promoted on an official level, especially in the case of Chinese medicine [159]. Conversely, sweet basil, being part of folk beliefs, rarely finds a place in school textbooks on religion. In the current official editions of school textbooks, after 75 years of discontinuance, *O. basilicum* is mentioned only once (in the edition for 5th grade, 10–12 years of age) in relation to the Feast of the Exaltation of the Holy Cross (14 September), when blessed sprigs are brought home after the liturgy [160]. It is completely missing in pre-Communism religion textbooks that comprised mostly Biblical excerpts, focusing on canonically recognized *Boswellia* sp. and *Commiphora* sp. resins, *Olea europaea* L. (oil and branches), *Triticum aestivum* L. (bread), and *Vitis vinifera* L. (wine) [161]. During Communism, *O. basilicum* was part of botanical education, presented as an ornamental plant and source of essential oils, together with other taxa of the Lamiaceae family [162]. The inclusion of traditional knowledge in education curricula could be challenging, especially when the formal systems favor “modern” data and/or methods of interpretation dissimilar from the local culture [163,164,165]. In Europe, where industrial agriculture and subsequent urbanization are underlying the reduction in direct experiences with nature, the creation of opportunities for more local, personal, and sensory experiences is seen as a leverage point in reshaping environmental education [166]. Logically, the introduction of local knowledge for such educational purposes would serve its revitalization, but in an updated form that reflects contemporary developments in science [167]. 

Modern pharmacology stems from traditional uses of medicinal plants, and based on current biomedical research, many biologically active compounds are clearly related to specific uses known for centuries as well as new ones [168]. Fast implementation of these new data allows the development of new medicines and functional foods that reach the market, and through advertisement and targeted promotion, supplies patients and consumers with transformed and upgraded traditional knowledge that they can combine with their own [167]. Screening of medicinal use of *O. basilicum* in the Balkans recorded during ethnopharmacological studies revealed quite varied, relatively low popularity (use value 0.17–0.3) that corresponds to frequencies of medicinal use of sweet basil reported by our participants [76,169,170,171]. Cultivated plants were frequently found to have higher use values than wild ones; however, ethnobotanical indices could be misleading due to the differences in methodologies and aims of the research [172,173].

The availability of new and/or foreign traditional knowledge, currently easily available, through different sources (i.e., literature and media) would allow for expanding overall interest and engagement with an already known (plant) species and ultimately enhance identification skills. On the other hand, commodification, especially of food plants, has set a specific perception frame that is not diversity-inclusive and contributes to (agro) biodiversity loss [174,175,176,177]. Hence, ethnobotanical data and research that targets various dimensions of ecosystem services and elucidates a richer knowledge should be more eagerly included in formal and informal educational agenda to attract more attention to plants and biodiversity as a whole [178,179,180]. Correspondingly, in this study, ten percent of the participants detailed their past use of *O. basilicum* bunches or individual sprigs also as a household repellent, placed in wardrobes and pantries, and one participant recalled it as being a pollinator attraction plant in vegetable gardens (Figure 9). Linalool, found in high concentrations in Bulgarian sweet basil varieties, is recognized as one of the important pollinator attractants as well as effective for plant protection against pests and fungal infections, hence suitable for organic farming and the adoption of agroecological practices [181,182,183,184]. Phenolic content was found to be related also to the microflora that different plant taxa harbor [185,186]. Sweet basil and other aromatic plants were reported in ethnographic sources as fermentation starters due to the beneficial lactic acid bacteria and yeast strains naturally occurring on them [187,188]. Currently, only one of our participants reported sweet basil as a leavening agent for the preparation of homemade bread, although they did not specify the usage of other ingredients. 

Association analyses of different uses of *O. basilicum* showed that cooking practices in the past were opposite to religious/ritual ones but correlated with its medicinal use (Table 5). 

Presently, the tendency is completely reversed, and all uses were found to be positively associated; however, food use had an insignificant association with the remaining uses. This implies that access to more varieties steered preference toward varieties suitable for the most frequent use—in the present study, food—with only 9.3% of CDC and 10.9% of CDD participants motivated to grow sweet basil for purposes entirely unrelated to food and/or other utilitarian purposes (Figure 10). 

A moderate association was present only between religious/ritual and medicinal uses and between ornamental and insect-repellent uses, which is in agreement with the concurrent importance of multipurpose sweet basil essential oil [107]. The traditional role of sweet basil in home gardens as an ornamental plant was preserved, but by few (14.4% of the participants), and the availability of new varieties was reflected in about twice as many participants currently appreciating its decorative side, but presumably not as essential for the selection. Ornamental value was found to be an important factor in the introduction of more plant taxa in home gardens, especially in rural areas where more gardening space is available [189,190]. However, in Bulgarian home gardens, food plants were found to be a leading incentive, a feature more typical of tropical home gardens [191,192,193], as they were used for subsistence farming before and during Communism [123]. Broadening the inquiries regarding preference and selection of varieties/landraces to grow and/or consume/utilize among a larger sample would elucidate motivation drivers to cultivate certain varieties/landraces.

## 3. Materials and Methods

Recent and historical ethnographic botanical and agricultural data were obtained by inquiries of major scientific databases (Web of Knowledge, Scopus, AGRIS, CAB Direct, and ERIH PLUS), as well as print-only sources available in public libraries. Previous ethnobotanical information on the cultivation and utilization of *O. basilicum* in Bulgaria was obtained during field studies of home gardens in 2017–2021 [76,123]. Image data were collected with the permission of the garden owners. An anonymous online questionnaire was constructed in Bulgarian using Google Forms format in a manner to avoid the collection of personal data of the participants; it was disseminated using personal contacts, mail lists, and social media outlets from the autumn of 2018 until December 2022. All participants were duly informed of the purpose of the research and their consent to participate was obtained before further questioning, following the guidelines prescribed in the Code of Ethics of the International Society of Ethnobiology [194]. Sociodemographic data were limited to age, sex, education, and information about their permanent residence in Bulgaria. Only adults were considered eligible for the research. Compliance was confirmed by the Scientific Council of the Institute of Biodiversity and Ecosystem Research, Bulgarian Academy of Sciences, acting as an independent institutional Ethics Board (Decision No. 6/21/05/21).

Online participants were asked to recognize images of six *O. basilicum* varieties/landraces together with one picture of *Mentha* sp. and another of *Salvia officinalis* (see Appendix A). Images were selected to represent the phenological stages (vegetative, flowering, and fruiting) and morphotypes of *O. basilicum* (3 foreign varieties and 3 landraces). *Mentha* sp. and *S. officinalis* were included as negative controls due to their common Bulgarian names, wild basil and horse basil, respectively. Further open and closed-end questions were related to:

Cultivation practices and motivation to cultivate sweet basil—in a garden and/or pots;

Current and past (childhood recollections) use in the following categories—food, religious/ritual, medicinal (including aromatic), insect repellent, ornamental, or other (open to additional uses);

Supposed “identity” of the presented sweet basil images related to the concept of “own” (Bulgarian) and foreign (Greek or Italian) and the means of participants to distinguish between them;

Seed saving and preference for planting material/seed origin.

### Statistical Analysis

Frequencies (absolute and percentages) were used to describe ordinal or nominal variables (sex, age groups, and education; number of correctly identified varieties/landraces and use categories). Studied variables were found to deviate from a normal distribution; therefore, we report the results of comparative non-parametric tests. Chi-Square tests were used to assess if the frequency distributions of the categorical variables were significantly different. The statistical association between nominal and ordinal variables was evaluated through Fisher’s exact test and Chi-Square correlation (*r*). The effect of sociodemographic parameters was initially assessed using the Mann–Whitney test (M-W) (between sexes), whereas the Kruskal–Wallis (K-W) test was applied for education levels and age ranges. Both tests produced insignificant results; therefore, we separated the participants into 2 groups according to the period they started their compulsory education (6–8 years of age): during Communism (1945–1989, CDC) and during Democracy (after 1989, CDD). The comparison between the two groups was sought due to the active role of schools in the indoctrination and prohibition of religious practices during Communism that would potentially affect traditional symbolic/religious use.

To examine the association between the identification skills, use categories, and participant preferences of the two groups (CDC and CDD), we estimated the Phi and Cramer’s V association coefficients φ_c_ (following Cohen’s guidelines [195], values were classified as 0.1 small, 0.3 medium, 0.5 large, in absolute value). All statistical tests were based on two-sided exact significance and with a significance level at least of α = 0.05. All analyses were performed using SPSS (ver. 20; IBM Corp., New York, NY, USA).

## 4. Conclusions

In modern (post-)industrial societies, “knowing your plant(s)” has increasingly become a “sudoskill” that laypeople often outsource to technology (devices and apps), together with the related issues and limitations [196]. A lack of immediate dependence on and/or attachment to certain (plant) species disengages a person’s attention and reduces a plant’s (positive) valence, especially when only a single use or function is known and/or prevalent. The preservation of cultural value, closer contact, and possession of some knowledge on the variety of services that a species and or an ecosystem provides is crucial for the recognition of its value and eventually its identity. Although the use frequencies and applications of crop plants are higher than those of wild ones, the reduced direct access to diverse characteristics limits identification skills. Further research on the effects of socioeconomic and cultural factors involving larger samples is needed, especially in countries where communities have undergone major political changes. The maintenance and promotion of a rich pool of varieties and landraces are vital both for the preservation of agrobiodiversity and the valuable cultural heritage that is under constant pressure from homogenization in the modern world. The halting of biodiversity loss and the restoration of ecosystems, aims of the United Nations in the current decade (until 2030), require a careful tailoring of measures that would enable more people to join proactively [197]. While conservation efforts are steered by international and national policies, the involvement and skills of the general public ensure the steady interest that would guarantee their implementation on the ground. Attaching diverse meanings and upholding local cultural values would prevent the commodification of resources and serve as a stepping stone to the development of more responsible attitudes toward nature awareness disparity [198]. As the effects of education and learning policies can be properly assessed only after several decades, it is important to urge policy makers and other stakeholders to adopt the most inclusive approaches to environmental knowledge. This will make it more accessible to the largest possible audience of any socioeconomic and cultural background.

## Figures and Tables

**Figure 1 plants-12-02771-f001:**
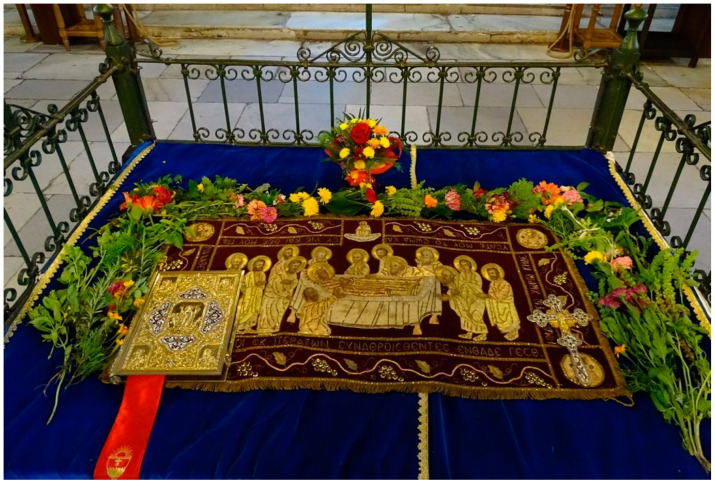
Church decoration for the Dormition of the Mother of God commemoration (courtesy of and with permission of Plovdiv Holy Metropoly [94]).

**Figure 2 plants-12-02771-f002:**
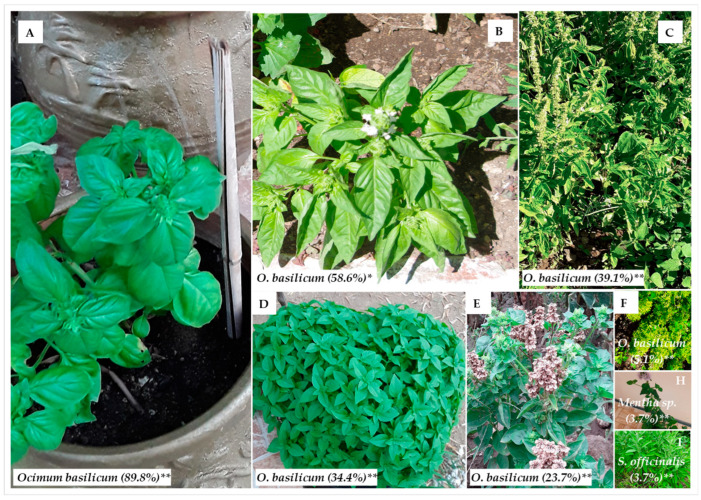
Treemap of participants (%, N = 215) recognizing “sweet basil” in the provided images: *Ocimum basilicum* “Genovese” variety (**A**); *O. basilicum* garden landrace from Ivaylovgrad (South Eastern Bulgaria, photo credit: Petar Petrov) (**B**); *O. basilicum* garden landrace from Samuilovo village (Southwestern Bulgaria) (**C**); *O. basilicum* “Bush” variety with small–medium ovate leaves (**D**); *O. basilicum* garden landrace, with fruits from Plevun village (Southeastern Bulgaria) (**E**); *O. basilicum* dwarf “Globe” variety (**F**); *Mentha* sp. (**H**); *Salvia officinalis* (**I**); * *p* < 0.05; ** *p* < 0.01 (Chi-Square test).

**Figure 3 plants-12-02771-f003:**
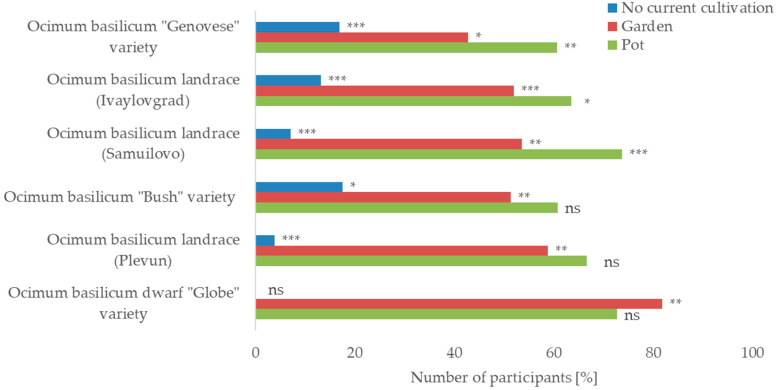
Effect of cultivation involvement on accuracy of *Ocimun basilicum* identification (N = 215); Fisher’s exact test: * *p* < 0.05, ** *p* < 0.01, *** *p* ≤ 0.001, ns—not significant.

**Figure 4 plants-12-02771-f004:**
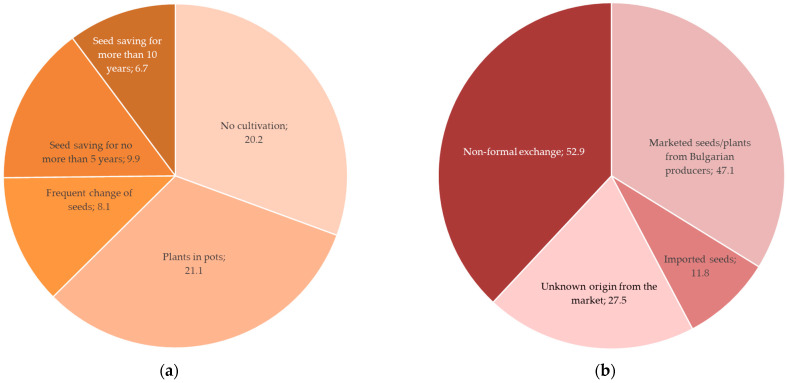
Procurement of *Ocimum basilicum* seeds and plants (%): (**a**) seed-saving practices (N = 147, *p* < 0.001, Chi-Square tests); (**b**) seed origin (N = 147).

**Figure 5 plants-12-02771-f005:**
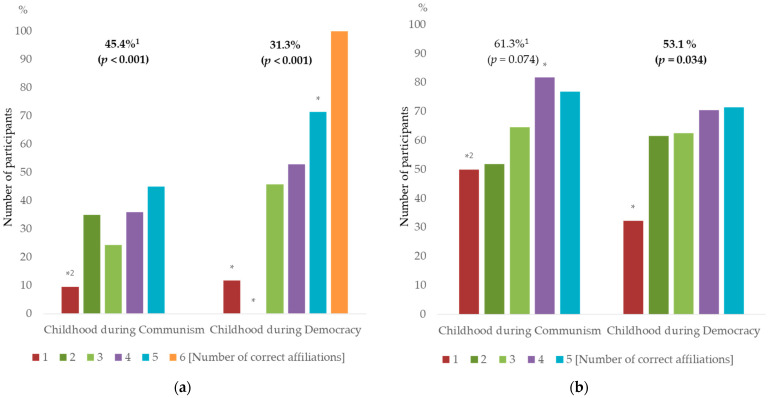
Number of correctly identified *Ocimum basilicum* varieties/landraces depending on interest in sweet basil cultivation in a garden (**a**) or in pots (**b**). ^1^ Percentage of participants involved in garden/pot cultivation (*p*-value, Chi-Square test). ^2^ Columns indicated with asterisk show significant difference between CDC and CDD groups involved or not in garden/pot cultivation (*p* < 0.05, Fisher’s exact test); N(CDC) = 115, N(CDD) = 100.

**Figure 6 plants-12-02771-f006:**
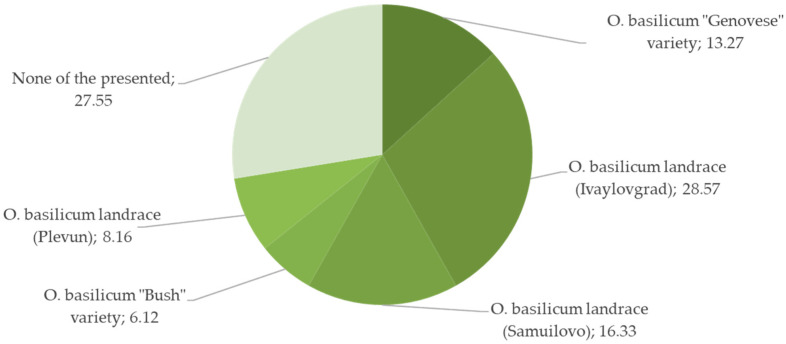
Affiliation of *Ocimun basilicum* images as “Bulgarian” basil (% of participants, N = 98).

**Figure 7 plants-12-02771-f007:**
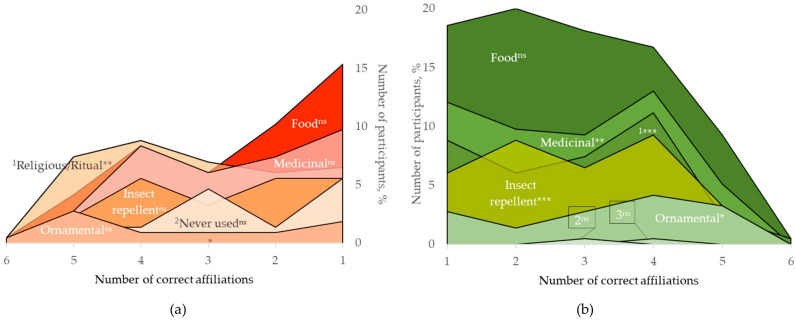
Frequencies of past (**a**) and current (**b**) uses of *Ocimum basilicum* by number of correctly affiliated images. ^1^ Religious/ritual use; ^2^ never used, ^3^ leavening agent; Fisher’s exact test: * *p* < 0.05, ** *p* < 0.01, *** *p* ≤ 0.001, ns—not significant; N = 215.

**Figure 8 plants-12-02771-f008:**
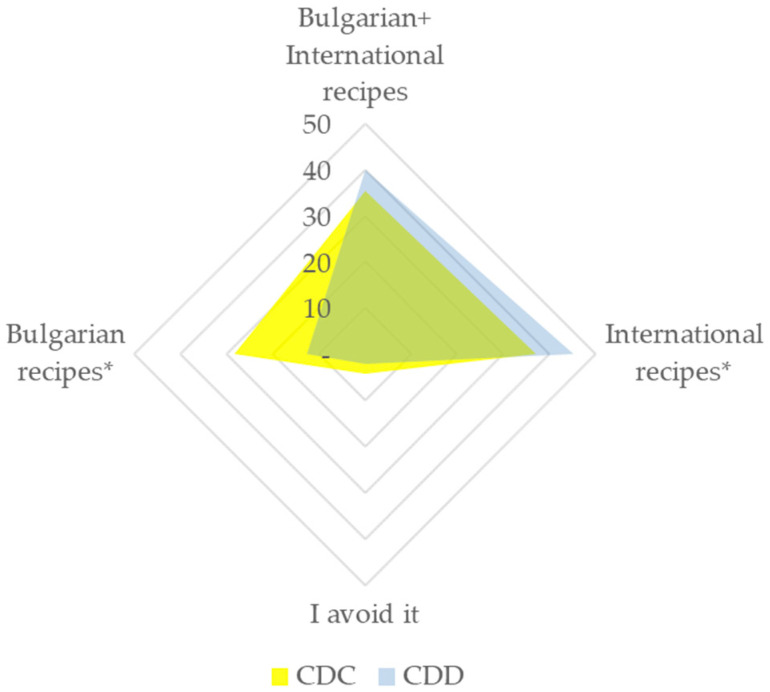
Preference of the participants (%) in consumption of *O. basilicum*. N(CDC) = 116; N (CDD) = 95; * significant difference between CDC and CDD groups (Fisher’s exact test).

**Figure 9 plants-12-02771-f009:**
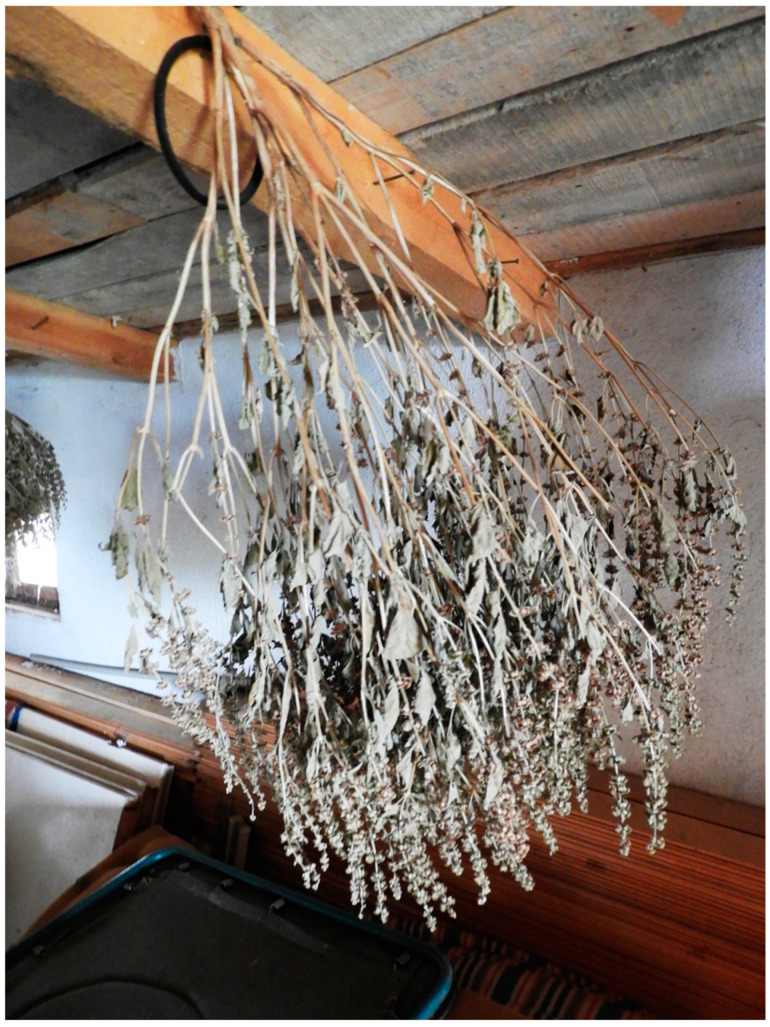
Dry *Ocimum basilicum* herbage, traditionally used as a household insect repellent, hung under the roof in the village of Plevun, southeastern Bulgaria.

**Figure 10 plants-12-02771-f010:**
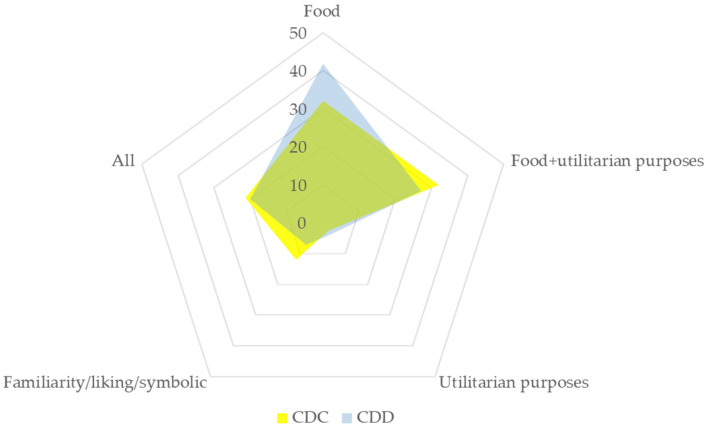
Motivation of participants (%) to grow *O. basilicum* at home and/or in a home garden. N(CDC) = 75; N (CDD) = 55; Fisher’s exact test *p* > 0.05.

**Table 1 plants-12-02771-t001:** Characteristics of the participants.

Parameter	Frequencies		
	Total	CDC ^1^	CDD
Age group (years)			
Under 20; N (%)	4 (1.9)	0 (0.0)	4 (4.0)
21–40; N (%)	96 (44.7)	0 (0.0)	96 (96.0)
41–60; N (%)	93 (43.3)	93 (80.9)	0 (0.0)
Over 60; N (%)	22 (10.2)	22 (19.1)	0 (0.0)
Total; N	215	115	100
Participant sex			
Female; N (%)	190 (86.0)	108 (90.8)	77 (80.2)
Male; N (%)	30 (14.0)	11 (9.2)	19 (19.8)
Total; N ^2^	215	119	96
Education level			
Primary; N (%)	1 (0.4)	0 (0.0)	1 (1.0)
Secondary; N (%)	39 (18.1)	25 (21.0)	14 (14.6)
College/University; N (%)	157 (73.0)	80 (67.2)	77 (80.2)
Non-disclosed; N (%)	18 (8.4)	14 (11.8)	4 (4.2)
Total; N	215	119	96
Biology/Agriculture proficiency and/or avid hobbyist; N (%)	38 (17.0)	19 (16.0)	18 (18.8)

^1^ Participants are grouped according to the time of their childhood (before start of the compulsory education) during or after the Communist era, 1945–1989. ^2^ Sample sizes vary due to missing data in the different variables. N: Sample size.

**Table 2 plants-12-02771-t002:** Accuracy of CDC and CDD participants in identification of *Ocimum basilicum*.

Code ^1^	Taxon	Setting, Number of Plants	Leaf Characteristics	Phenophase	CDC (%)	CDD (%)	Fisher’s Exact Test (*p*, 2-Saided)	Phi and Cramer’s V, φ_c_
A	*Ocimum basilicum* “Genovese” variety	potted, single individual	large/medium elliptic	Vegetative	52.1	37.7	0.024	0.16 (small)
B	*Ocimum basilicum* landrace (Ivaylovgrad)	garden, single individual	medium–large/medium ovate	Flowering	30.2	28.4	>0.05	0.09 (NS)
C	*Ocimum basilicum* landrace (Samuilovo)	garden, multiple individuals	medium–large/medium ovate	Flowering	22.3	16.7	>0.05	0.029 (NS)
D	*Ocimum basilicum* “Bush” variety	potted, single individual	small–medium/ovate	Vegetative	20.0	14.4	>0.05	0.04 (NS)
E	*Ocimum basilicum* landrace (Plevun)	garden, single individual	medium–large/ovate-elliptic	Fruiting	11.6	12.1	>0.05	0.071 (NS)
F	*Ocimum basilicum* dwarf “Globe” variety	garden, multiple individuals	very small/narrow elliptic	Vegetative	2.3	2.8	>0.05	0.046 (NS)
H	*Salvia officinalis*	garden, multiple individuals	medium/oblong–lanceolate	Vegetative	53.0	43.3	>0.05	0.028 (NS)
I	*Mentha* sp.	potted, single individual	small–medium/elliptic	Flowering	52.1	44.2	>0.05	0.124 (small)

^1^ Code letters follow those shown in Figure 2; N(CDC) = 115, N(CDD) = 100; CDC—Childhood during Communism, CDD—Childhood during Democracy; NS—non-significant.

**Table 3 plants-12-02771-t003:** Association between past and present uses of *Ocimum basilicum* in Bulgaria (*r*, Chi-Square Correlation).

Past	Food	Religious/ Ritual	Medicinal	Insect Repellent	Ornamental
Current					
Food	0.073	0.011	−0.053	−0.044	**−0.152 ***
Religious/Ritual	**−0.175 ****	**0.576 ****	0.002	**0.165 ***	0.028
Medicinal	−0.112	**0.208 ****	**0.340 ****	**0.157 ***	−0.005
Insect Repellent	0.095	0.119	0.117	0.076	−0.073
Ornamental	−0.005	0.128	0.075	0.105	**0.151 ***

Data in bold are significant at * *p* < 0.05, ** *p* < 0.01.

**Table 4 plants-12-02771-t004:** Present and recalled past uses of *Ocimum basilicum* in Bulgaria by CDC and CDD participants (number of participants who gave positive answers).

Time of Use	Past					Current		
Use Category	CDC	CDD	Phi and Cramer’s V	Fisher’s Exact Test (*p*, 2-Sided)	CDC	CDD	Phi and Cramer’s V	Fisher’s Exact Test (*p*, 2-Sided)
Food	44	49	**0.141**	0.05	112	91	0.015	1.00
Religious/Ritual	44	20	**0.175**	0.011	52	25	**0.183**	0.01
Medicinal	38	34	0.037	0.663	67	37	**0.177**	0.013
Insect Repellent	29	18	0.068	0.407	37	35	0.057	0.468
Ornamental	11	4	0.099	0.183	21	10	0.102	0.172

Data in bold are significant at *p* < 0.05; Phi and Cramer’s V association coefficient: > 0.1 (small); 0.3 (medium); 0.5 (large).

**Table 5 plants-12-02771-t005:** Association between different uses of *Ocimum basilicum* in Bulgaria (*r*, Chi-Square correlation).

Past	Religious/ Ritual	Medicinal	Insect Repellent	Ornamental
Food	**−0.177 ****	**0.144 ***	0.078	0.058
Religious/Ritual		0.03	0.127	−0.061
Medicinal			**0.187 ****	0.001
Insect Repellent				−0.01
**Current**				
Food	0.035	0.024	0.023	0.051
Religious/Ritual		**0.337 ****	**0.290 ****	**0.222 ****
Medicinal			**0.274 ****	**0.292 ****
Insect Repellent				**0.300 ****

Data in bold are significant at * *p* < 0.05, ** *p* < 0.01.

## Data Availability

Research data contains private data of the questionnaire participants and could be provided upon request, only if ethical and ethical restrictions are met.

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
