# Peer review of "Sweet Basil between the Soul and the Table—Transformation of Traditional Knowledge on *Ocimum basilicum* L. in Bulgaria"

_plants, 2023, doi:10.3390/plants12152771_

Round 1
Reviewer 1 Report
Dear Editor, dear Authors,
I've completed my review of the manuscript entitled Sweet basil between the soul and the table – Transformation of traditional knowledge on Ocimum basilicum L. in Bulgaria , authors: Teodora Ivanova 1,*, Yulia Bosseva 1, Mihail Chervenkov 1,2 and Dessislava Dimitrova
The authors present original results of their field and laboratory/analysys work; in focus is ritual, ornamental, and medicinal plant Ocimum basilicum which traditional usage is nowadays in knowledge transformation.
Although previous data for Bulgarian rural home gardens showed significant number of preserved local landraces, it remained unclear how people perceive large varietal diversity of this species and how the traditions related to its use are preserved.
Therefore, authors combined a literature review on the cultural value of sweet basil and the breeding of local genetic resources with an online questionnaire, that sought to access different aspects of past (recalled) and present use and related knowledge. Identification skills of the participants were tested using images of local plant landraces and foreign varieties.
Authors consult really a lot of reference data (181), which are mostly recent (within 5 years), which is highly commendable. Introduction is very nice, clear and well written, taking into account all aspects of the investigated problem. Methods are adequately chosen, and statistically fully processed, following the guidelines prescribed in the Code of Ethics of the International Society of Ethnobiology.
Prior ethnobotanical data on the cultivation and utilization of O. basilicum in Bulgaria was obtained during own field studies of home gardens in 2017-2021.
The results showed flowering sweet basil is used for decoration of sacred breads and icons together with other seasonal ornamentals like Geranium macrorrhizum L., Chrysanthemum indicum L., Zinnia elegans Jacq., Tagetes erecta L., etc. Sweet basil oil is also used as non-essential ingredient of the Great Chrism made by Bulgarian and Romanian Orthodox Churches. Also, medicinal properties of sweet basil are quite popular in the Bulgarian folk and modern medicine. The flowering herbage and seeds are mentioned in numerous preparations in all major professionally written phytotherapy textbooks and herbals.
Participants who spent their formative years and started their compulsory education (6-8 years of age) before the end of Communism in Bulgaria were 51% of the sample (coded hereafter CDC – Childhood during Communism as opposed to CDD-Childhood during Democracy sub-set).
Interesting was the absolute number of CDC participants who identified presented taxa was higher than of CDD, and only for one variety and one landrace the difference was in favor of CDD, by only one participant. When comparing the accuracy of CDC and CDD groups, there was significant difference only for the “Genovese” variety (p = 0.024, Fisher's exact test). Identification skills and the time of the participant’s childhood were significantly associated only for the “Genovese” variety and Mentha sp. (φc > 1, small association). Various factors were shown previously to influence plant identification skills, i.e., local plant richness, demographic and socioeconomic factors, proficiency/professional involvement, source(s) of information, personal interests, etc.
In results is presented, all participants were reluctant to assign specific use to any of the three country-bound identities, which was opposite to data on Italian diaspora in Romania who discriminate between “their” (Romanian) and “our” (Italian) sweet basil solely according to its use, ritual and culinary, respectively. However, further exploration of the identification – usage nexus showed that high positive identification of the “Genovese” variety was reflected in the use frequencies.
Authors presented this study on very interesting way connecting old and new knowledge: Ocimum basilicum was most frequently reported as food than as ritual/symbolic plant which was preserved but devalued during the Communism regime (1945-1989). Food and religious uses were negatively associated in the past, but presently the tendency is completely reversed.
The results are presented concisely, accurately and according to the botanical nomenclature rules. Researched are all relevant sources.The figures/tables/images/schemes are appropriate and properly show the data, they are easy to interpret and understand.
Discussion is very well done, the authors results revealed a sharp distinction between Ocimum basilicum and other similar Lamiaceae species that are represent in research.
I don’t have other comments on this part. I find the presented results here important, because provide the essential information about preservation of cultural value, closer contact and possession of some knowledge on the variety of services that a species and or an ecosystem provides is crucial or the recognizing their value and eventually their identity. Therefore, such background information should be published to become accessible for broader audience. Also, maintenance and promotion of rich pool of varieties and landraces is vital not only for preservation of agrobiodiversity but also, by matching expectations and needs of more users/consumers,contributes to preserving the access to valuable cultural heritage that is constantly pressured by homogenization in the modern world.
There are just a few minor comments; in section References (need to finish the reference 121, 122, 128).
The paper is written very clear and concise; results are suitable for the publication in MDPI Plants. I herewith recommend the reviewed manuscript for publication in the Plants MDPI.
SPECIFIC COMMENTS:
Specific comments are listed in the attached word file

Author Response
Dear Reviewer,
Dear Editors,
Thank you very much for your encouragement to our attempt to explore the transformation of traditional knowledge on multipurpose economic plants from an ethnobotanical point of view using Bulgarian experiences with sweet basil as an example. We are grateful for your suggestions how to improve the manuscript. We took them in consideration during the preparation of the revised version of the manuscript. You can see all changes and additions in the provided file with track changes.

Reviewer 2 Report
The paper presents an interesting study on the use in the past and present and on the ability to identify Ocimum basilicum varieties by adult laypeople. The study is based on information available in the literature and on an online questionnaire. The research is competently conducted, and a large number of references were used for the literature review. Yet, I have some comments, as follows:
General:
-Some improvements in the fluency of the language are desirable.
-I suggest that the long Results and Discussion section would be split into separate Results and Discussions sections.
Specific:
-Introduction, lines 49-50: Is the mentioned geographical emphasis really correct, i.e., not including, for instance, the rest of Europe?
-Introduction: It would be good to provide more information about the conservation actions, gene bank collections, etc. of Ocimum basilicum.
-Questionnaire: The demographic information of the subjects is given. However, how was the recruiting performed? It seems that many of them were involved in cultivating Ocimum (more than the average population). Whether having schooling under communism or democracy is strongly connected with age (young subjects having schooling under democracy). Was it possible to distinguish the regime effect reliably from the age effect? How were the answering/identification circumstances organized? Obviously online, but was there a time limit or possibility to use other resources to help identification?
-Conclusions, lines 510-511: It is not true that breeding programs could not favor many desirable traits in a single variety. Traits can be combined through breeding.
-Discussion/Conclusions:
--It should be mentioned that the conservation of genetic resources and agrobiodiversity does not rely only on individuals and home gardens, but national/international conservation actions and gene banks have a crucial role in conservation, characterization, and exploitation of genetic resources. Overall, developments in organizational conservation actions of genetic resources have been improving.
--It would be good to describe more what those changes in education and life were when moving from communism to democracy, in this case in Bulgaria.
-Conclusions “Further research on the effects of socioeconomic and cultural factors involving larger samples is needed, especially in countries where communities have undergone major political changes”. This is true. Yet, there are other changes in societies from the “old” to “modern times” that have affected the use and knowledge of traditional plants and are of interest to be investigated. This statement could be widened in Discussion/Conclusions to go beyond the transition from communism to democracy.
Some proofreading/editing recommended.
Author Response
Dear Reviewer,
Dear Editors,
Thank you very much for your valuable suggestions to our efforts to explore the transformation of traditional knowledge on sweet basil in Bulgaria. We are grateful for your critical notes and suggestions how to improve the manuscript. We took them in consideration during the preparation of the revised version of the manuscript. You can see all changes and additions in the provided file with track changes.
Please find below our answers (in bold) to reviewer’s specific comments to the text:
- I suggest that the long Results and Discussion section would be split into separate Results and Discussions sections.
Thank you for this remark. We opted for combined Results and Discussion section because data from the literature review and the online questionnaire was quite diverse. We considered that it would be more convenient for the readers to track different aspects presented in the Results along with matching discussion and to avoid repetitions and transitions in separate Discussion section.
- Some improvements in the fluency of the language are desirable.
Thank you for this comment. We made thorough re-check of the English language and made corrections where needed. (see file with corrections made).
- Introduction, lines 49-50: Is the mentioned geographical emphasis really correct, i.e., not including, for instance, the rest of Europe?
Thank you for this correction, we revised the text accordingly (see lines 53-54).
- Introduction: It would be good to provide more information about the conservation actions, gene bank collections, etc. of Ocimum basilicum.
Following information was added in the Introduction: Sweet basil was reported recently for landrace hotspots in Italy and Greece, however, about half of the accessions in European gene banks (total of 816) are stored Germany, Croatia and Ukraine [51, 72].
- Questionnaire: The demographic information of the subjects is given. However, how was the recruiting performed? It seems that many of them were involved in cultivating Ocimum (more than the average population).
Thank you for this question. As we mention in the Materials and Methods participants were recruited by disseminating the questionnaire using personal contacts, mail lists and social media outlets (lines 463-4). Maintenance of kitchen gardens with various herbs is very popular in Bulgaria, and even in the urban and peri-urban areas vegetable plots and greenhouses are preferred way of use of home gardens and yards (see https://www.youtube.com/watch?v=VKqEmkkElsg, 25 min onwards for some visuals). However, extended research would be needed to access current rates in the general population as Bulgarian state has no register nor statistics devoted to home garden spaces. Additionally, non-formal seed exchange and seed saving would misleadingly reassure lay-people in the authenticity of the landraces and varieties as the opportunity for cross-pollination and collection of hybrid seeds is also highly possible (see added text 262 onwards)
- Whether having schooling under communism or democracy is strongly connected with age (young subjects having schooling under democracy). Was it possible to distinguish the regime effect reliably from the age effect?
Thank you for this remark. Age or sex were found non-significant for plant identification, when study sample was taken as a whole (lines 221-223). We decided to split the sample according the start of the formal education as schools were one of main places for Communist propaganda for devaluation of religion and instilment of anti-Christian sentiments, including reprimands for participation in rituals in public and temple visitations (Petrov, 2000). These processes were most active between the 1960s-1980s, when most of the older participants were born. Between 1956-1985 there was no religion census as the state considered Bulgarian population as “antireligious” (https://www.nsi.bg/Census/StrReligion.htm ), but in 2001 nearly 6.5 million Bulgarians were considered Orthodox Christians, similar to the numbers after WWII. Currently, little over 4 million are identifying themselves as Orthodox Christians (https://www.nsi.bg/en/content/19874/%D0%BF%D1%80%D0%B5%D1%81%D1%81%D1%8A%D0%BE%D0%B1%D1%89%D0%B5%D0%BD%D0%B8%D0%B5/ethno-cultural-characteristics-population-september-7-2021 ) so we decided to explore the difference between these two groups as interest to larger varietal diversity was expected to be connected to ritual and the related ornamental use. Additional research would be needed so to discern the effects of industrialization/modernization as they coincide with the onset of the Communist regime and abandonment of typical rural livelihoods Bulgarians due to the expropriation of private land so it is outside the scope of our research and expertise.
- How were the answering identification circumstances organized? Obviously online, but was there a time limit or possibility to use other resources to help identification?
Thank you for this interesting question. The participants were not limited in the time they could spend for the identification and the only limitation was that they could not correct their responses after submission. Naturally, online questionnaires rely on the participants’ willingness to provide genuine information. Given the fact that accuracy was high only for two of the presented 6 sweet basil varieties/landraces it is questionable if participants have used some online identification apps or other help for the identification. It would be curious to add suitable questions (in future studies) that would reveal if participants have used such assistance.
- Conclusions, lines 510-511: It is not true that breeding programs could not favor many desirable traits in a single variety. Traits can be combined through breeding.
It should be mentioned that the conservation of genetic resources and agrobiodiversity does not rely only on individuals and home gardens, but national/international conservation actions and gene banks have a crucial role in conservation, characterization, and exploitation of genetic resources. Overall, developments in organizational conservation actions of genetic resources have been improving.
It would be good to describe more what those changes in education and life were when moving from communism to democracy, in this case in Bulgaria.
Conclusions “Further research on the effects of socioeconomic and cultural factors involving larger samples is needed, especially in countries where communities have undergone major political changes”. This is true. Yet, there are other changes in societies from the “old” to “modern times” that have affected the use and knowledge of traditional plants and are of interest to be investigated. This statement could be widened in Discussion/Conclusions to go beyond the transition from communism to democracy.
We appreciate these remarks. Our statement was regarding breeding of varieties that would be suitable for multipurpose use but in some cases, like in sweet basil, in example traits sought for culinary use (with large tender leaves) are in contradiction with insect repelling ones (i.e., high linalool content) or larger inflorescences for the ritual/ornamental use. The conclusion was revised and text on use of traditional knowledge and how sweet basil was represented in Bulgarian school textbooks was added to the Results and Discussion in this regard (line 389 onwards).

Round 2
Reviewer 2 Report
The revision was well prepared.
Author Response
Dear Reviewer,
After considering the suggestions from Editors we propose following changes:
In the Results and Discussion (lines 390-411)
Numerous books published in Bulgaria that cite folk medicine indicated that traditional knowledge was not neglected but favored by the state during the Communism, in contrast to religion. Hence, the interest not only to traditional / ”old” knowledge but also traditional medicine of other cultures was promoted on an official level, especially in the case of Chinese medicine [159]. Conversely, the sweet basil, being part of the folk believes, rarely finds place in religion school textbooks. In the current official editions of school textbooks, after 75 years of discontinuance, O. basilicum is mentioned only once (in the edition for 5th grade, 10-12 years of age) in relation to Feast of the Exaltation of the Holy Cross (14 September), when blessed sprigs are brought home after the liturgy [160]. It is completely missing in pre-Communism religion textbooks that were comprised mostly of Biblical excerpts, focusing on canonically recognized Boswellia sp. and Commiphora sp. resins, Olea europaea L. (oil and branches), Triticum aestivum L. (bread) and Vitis vinifera L. (wine) [161]. During the Communism O. basilicum was part of the botanical education, presented as ornamental plant and source of essential oils, together with other taxa of Lamiaceae family [162]. Inclusion of traditional knowledge in education curricula could be challenging, especially when the formal systems favor “modern” data and/or methods of interpretation dissimilar with the local culture [163–165]. In Europe, where industrial agriculture and subsequent urbanization are underlying for the reduced direct nature experiences, creation of opportunities for more local, personal and sensory experiences are seen as leverage point in reshaping environmental education [166]. Logically, introduction of local knowledge for such educational purposes would serve for its revitalization, yet, in an updated form that reflects contemporary developments in science [167].
In the Conclusion (lines 555-569)
Maintenance and promotion of rich pool of varieties and landraces is vital both for preservation of agrobiodiversity and the valuable cultural heritage that is under constant pressure by the homogenization in the modern world. Halting of biodiversity loss and restoration of ecosystems, that is aimed by United Nations in the current decade (untill 2030) requires careful tailoring of measures that would enable more people to join proactively [197]. While conservation efforts are steered by international and national policies, the involvement and skills of the general public ensures the steady interest that would guarantee their implementation on the ground. Attaching diverse meanings and upholding local cultural values would prevent commodification of resources and serve as a stepping stone to the creation of more responsible attitudes against nature awareness disparity [198]. As far as the effects of education and learning policies could be properly assessed only after several decades, it is important to urge policy makers and other stakeholders to adopt most inclusive approaches to environmental knowledge. This will make it more accessible for largest possible audience of any socioeconomic and cultural background.